# A shift in glutamine nitrogen metabolism contributes to the malignant progression of cancer

Manabu Kodama[1], Kiyotaka Oshikawa[1], Hideyuki Shimizu[1], Susumu Yoshioka[1,2], Masatomo Takahashi [3], Yoshihiro Izumi [3], Takeshi Bamba [3], Chisa Tateishi[1], Takeshi Tomonaga[4], Masaki Matsumoto [5✉] & Keiichi I. Nakayama[1,5✉]

Glucose metabolism is remodeled in cancer, but the global pattern of cancer-specific metabolic changes remains unclear. Here we show, using the comprehensive measurement of metabolic enzymes by large-scale targeted proteomics, that the metabolism both carbon and nitrogen is altered during the malignant progression of cancer. The fate of glutamine nitrogen is shifted from the anaplerotic pathway into the TCA cycle to nucleotide bio-synthesis, with this shift being controlled by glutaminase (GLS1) and phosphoribosyl pyr-ophosphate amidotransferase (PPAT). Interventions to reduce the PPAT/GLS1 ratio suppresses tumor growth of many types of cancer. A meta-analysis reveals that PPAT shows the strongest correlation with malignancy among all metabolic enzymes, in particular in neuroendocrine cancer including small cell lung cancer (SCLC). PPAT depletion suppresses the growth of SCLC lines. A shift in glutamine fate may thus be required for malignant progression of cancer, with modulation of nitrogen metabolism being a potential approach to SCLC treatment.

[1] Department of Molecular and Cellular Biology, Medical Institute of Bioregulation, Kyushu University, 3-1-1 Maidashi, Higashi-ku, Fukuoka, Fukuoka 812-8582, Japan. [2] LSI Medience Corporation, 1-13-4 Uchikanda, Chiyoda-ku, Tokyo 101-8517, Japan. [3] Division of Metabolomics, Medical Institute of Bioregulation, Kyushu University, 3-1-1 Maidashi, Higashi-ku, Fukuoka, Fukuoka 812-8582, Japan. [4] Laboratory of Proteome Research, National Institute of Biomedical Innovation, Health, and Nutrition, 7-6-8 Saito-Asagi, Ibaraki, Osaka 567-0085, Japan. [5] Division of Proteomics, Medical Institute of Bioregulation, Kyushu University, 3-1-1 Maidashi, Higashi-ku, Fukuoka, Fukuoka 812-8582, Japan. ✉email: masakim@bioreg.kyushu-u.ac.jp; nakayak1@bioreg.kyushu-u.ac.jp

Glucose and glutamine are two nutrients that support energy production and biomass synthesis in the cell. Glucose is a major source of carbon not only for ATP production via the tricarboxylic acid (TCA) cycle, which is linked to oxidative phosphorylation in mitochondria, but also for the generation of precursors of nucleotides, amino acids, and fatty acids. Glutamine contributes to the supply of not only carbon but also nitrogen, the latter of which is required for biosynthesis of diverse compounds such as purine and pyrimidine nucleotides[1], glucosamine 6-phosphate, and nonessential amino acids[2–4]. Glutamine is thus rate-limiting for cell cycle progression, with glutamine deprivation arresting the cell cycle in S phase in certain cellular contexts[5,6].

Cancer tissue consumes much more glucose than does non-proliferating normal tissue, with most of this glucose being converted to lactate even in the presence of oxygen. This metabolic shift from oxidative phosphorylation to glycolysis in cancer is known as the Warburg effect[7]. Glutamine also serves as an important source of carbon for cellular bioenergetic and biosynthetic needs in cancer. The conversion of glutamine to α-ketoglutarate (α-KG) via glutamate serves as a route for entry of glutamine-derived carbon into the TCA cycle. Cleavage of the γ-nitrogen of glutamine by glutaminase (GLS) generates glutamate and free ammonia in this process and is a rate-limiting step of the glutamine anaplerosis. GLS1 is frequently dysregulated in cancer[8–10], with its expression having been shown to be positively regulated by c-Myc[11]. Several GLS inhibitors have been developed and have shown tumor-suppressive activities in preclinical models[8,12–14], although such agents have been found to lack therapeutic effects in certain instances[15,16]. The role of GLS in cancer development appears to be highly context dependent and remains unclear[17,18].

In addition to serving as a carbon source, glutamine is important for cancer growth as a nitrogen donor, in which capacity it supports the increased demand for nucleotide biosynthesis in cancer cells[2]. The increased utilization of glutamine nitrogen in nucleotide production is facilitated by growth-promoting signals. For instance, elevated levels of c-Myc induce the expression of several enzymes in nucleotide biosynthetic pathways, including phosphoribosyl pyrophosphate amido-transferase (PPAT), which transfers the γ-nitrogen of glutamine to 5-phosphoribosyl pyrophosphate (PRPP), a key rate-limiting reaction in purine biosynthesis[19,20]. Indeed, the expression of PPAT was shown to be increased in lung adenocarcinoma and related to disease prognosis in patients[21]. Although glutamine is directly subjected to catabolic and anabolic reactions by GLS1 and PPAT, respectively, the relation between the fate of glutamine and cancer malignancy has remained to be fully elucidated.

An understanding of the global network of metabolic reactions will require precise and comprehensive measurement of a large number of proteins. We have recently developed a large-scale targeted proteomics platform, designated iMPAQT (in vitro proteome-assisted MRM for protein absolute quantification), that allows absolute quantification of all metabolic enzymes[22]. We have now taken advantage of iMPAQT in association with metabolomics analysis with stable isotope-labeled glutamine to provide a global view of cancer metabolism. We have thereby uncovered a mechanism by which the fate of nitrogen in gluta-mine is shifted from the glutamine anaplerotic pathway into the TCA cycle to nucleotide biosynthesis during malignant progression of cancer. Most importantly, a meta-analysis integrating many cancer cohorts that comprise ~11,000 patients indicated that the enzymes that contribute to this nitrogen shift are decisive prognostic markers. In particular, PPAT expression is one of the strongest indicators for poor prognosis in small cell lung cancer (SCLC), which is highly malignant (5-year survival rate, ~7%),

with a high rate of growth and metastasis, and has been referred to as a "recalcitrant cancer"[23], with no specific and effective treatment having been developed to date. Our results suggest that intervention in glutamine metabolism is a promising approach to the establishment of cancer treatments, especially for SCLC.

## Results

**Establishment of a malignant progression model**. An overview of the study is shown in Fig. 1a. To delineate the global landscape of the shift in metabolism during malignant progression of cancer with the iMPAQT system, we first established a model for malignant transformation. Normal human diploid fibroblast (TIG-3) cells were infected with retroviruses containing the early region of SV40 and encoding the catalytic component of human telomerase (hTERT) and human c-Myc. The resultant trans-formed cells (TSM cells) gained a moderate ability to undergo anchorage-independent growth[22,24] (Fig. 1b, c). TSM cells were essentially unable to form tumors on transplantation into nude mice (Fig. 1d), and they therefore could be considered to be premalignant. Given that the ability of cells to undergo anchorage-independent growth is correlated with that to form tumors in nude mice[25], TSM clones grown in three-dimensional (3-D) culture were picked up and expanded in 2-D culture to obtain AIG (anchorage-independent growth)–1 clones. This selection process was repeated three times to yield AIG-3 cells, which showed a pronounced ability to form colonies in 3-D culture as well as to form tumors on transplantation into nude mice (Fig. 1b–d). The malignant progression from TSM to AIG-3 was associated with a stepwise increase in the expression of c-Myc (Fig. 1e, f).

**Global view of proteomic and metabolomic changes**. We applied iMPAQT to TSM as well as AIG-1, -2, and -3 clones in order to comprehensively measure the abundance of 342 metabolic enzymes. The amounts of certain glycolytic enzymes including hexokinase 2 (HK2), enolase 1 (ENO1), and lactate dehydrogenase A (LDHA) were increased in AIG-3 cells compared with TSM cells, with glucose uptake also being increased (Fig. 2a and Supplementary Fig. 1a). These results suggested that the Warburg effect is more pronounced in AIG-3 cells than in TSM cells. Gene Ontology (GO) enrichment analysis revealed that the most affected biological process in AIG-3 cells compared with TSM cells was nucleic acid biosynthesis (Fig. 2b). The amounts of most enzymes in nucleotide biosynthesis pathways (both de novo and salvage pathways) including PPAT, which metabolizes glutamine, were thus cooperatively increased in AIG-3 cells compared with TSM cells, whereas the expression of GLS1, the rate-limiting enzyme of the glutamine anaplerotic pathway, was downregulated in AIG-3 cells (Fig. 2c, d). The pyruvate dehydrogenase E1 alpha subunit (PDHA1), which oxidizes pyr-uvate for TCA anaplerosis, was upregulated in AIG-3 cells (Fig. 2c, d), consistent with a previous finding[16]. To confirm these changes in glucose metabolism observed at the proteome level, we performed mass isotopomer analysis with [$^{13}C_6$]glucose[16] (Fig. 2e), and we found that both the $^{13}C$-labeling efficiency and metabolite pool size for citrate, aspartate, and glutamate were markedly increased in AIG-3 cells compared with TSM cells (Fig. 2f–h and Supplementary Fig. 1b–d), consistent with the results of previous studies[3,16,26]. In particular, the increase in the M + 2 fraction of citrate, aspartate, and glutamate was consistent with the notion that carbon sources from pyruvate mediated by PDHA1 was increased in AIG-3 cells compared with TSM cells.

Our results thus suggested that the amounts of PPAT in the de novo nucleotide biosynthesis pathway and of GLS1 in the glutamine anaplerotic pathway into the TCA cycle, two enzymes

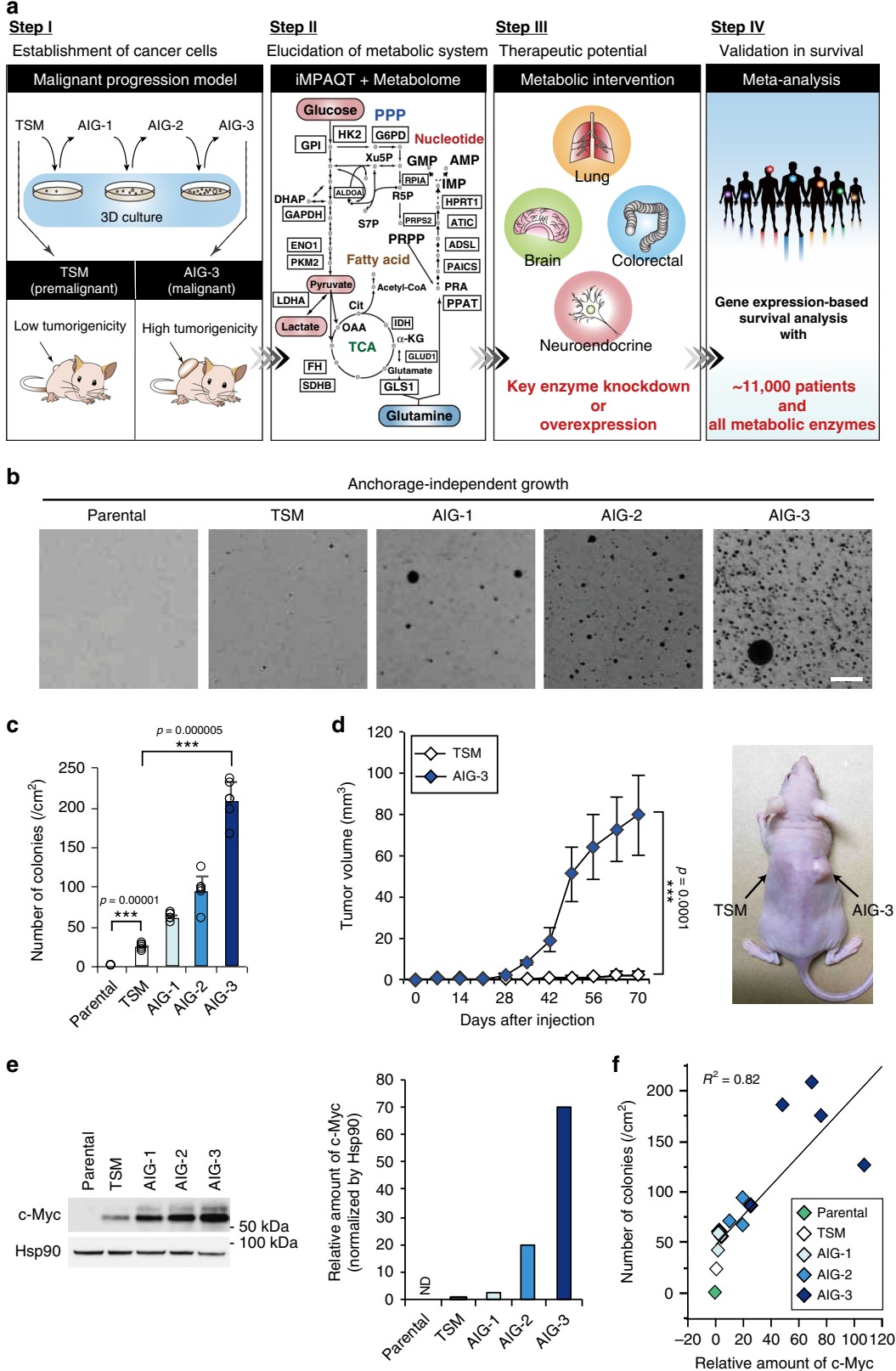

that rely on glutamine as a key substrate, are changed in a reciprocal manner during malignant progression. Similar to the genes for cyclin D2 (*CCND2*) and LDHA, expression of the PPAT gene is regulated by c-Myc[20,27,28] and was found to be increased in AIG-3 cells (Supplementary Fig. 1e), consistent with the observation that c-Myc is highly upregulated in these cells

(Fig. 1e). In contrast, the amount of GLS1 mRNA was downregulated in AIG-3 cells (Supplementary Fig. 1e). The abundance of GLS1 was previously shown to be indirectly controlled by c-Myc through negative regulation of the micro-RNA mir-23a, which suppresses expression of the GLS1 gene[11]. However, another study found that c-Myc transactivates the mir-

**Fig. 1 Establishment of a malignant progression model. a** Experimental flow. We first established transformed cells by introduction of hTERT, the SV40 early region, and c-Myc (TSM) into the normal human diploid fibroblast line TIG-3. To recapitulate stepwise malignancy, we serially selected cells grown in 2.5% methylcellulose culture and obtained clones (designated AIG) with an increased ability to undergo anchorage-independent growth. We next applied iMPAQT and metabolomics analyses to the malignant progression model, with the resulting findings being subjected to validation with an integrated meta-analysis. Icon made by Pixel perfect from www.flaticon.com (https://www.flaticon.com/free-icon/standing-human-body-silhouette_30473). **b** Phase-contrast microscopy of colonies stained with 2-(4-iodophenyl)-3-(4-nitrophenyl)-5-phenyl-2*H*-tetrazolium chloride (INT) showing anchorage-independent growth in the malignant progression model. Colony formation assay were conducted with three biological replicates ($n = 3$). Scale bar, 2 mm. **c** Anchorage-independent growth was quantitated by counting of colonies stained with INT. Colonies were counted in two randomly selected fields in each of three dishes. ($n = 6$). **d** Volume of tumors formed by TSM or AIG-3 cells after subcutaneous injection into female athymic nude mice ($n = 6$). **e** Immunoblot analysis of c-Myc and Hsp90 (loading control) in the malignant progression model, was conducted with single time. The intensity of the c-Myc band was measured by densitometry. ND, not detected. **f** Scatter plot showing the correlation of c-Myc abundance and anchorage-independent growth in the malignant progression model. Results for several different AIG-1, -2, or -3 clones are shown. $R^2$, correlation coefficient. Data in **c** and **d** are means ± s.d. ***$P$ < 0.001 (paired two-tailed Student's $t$-test). Source data are provided as a Source Data file.

23a gene[29]. Indeed, previous studies have shown that GLS1 expression was not increased or was even decreased by overexpression of c-Myc[30,31]. Despite our observation that c-Myc was highly upregulated in AIG-3 cells, the abundance of mir-23a was also substantially increased in these cells compared with TSM cells (Supplementary Fig. 1f), consistent with the reduced abundance of GLS1 mRNA and protein.

To confirm the changes in glutamine metabolism observed at the proteome level, we performed mass isotopomer analysis[32] with [$^{13}C_5$/$^{15}N_2$]glutamine, in which all carbon and nitrogen atoms are replaced with stable isotopes, and with TSM and AIG-3 cells (Fig. 3a). With the use of Orbitrap-based mass spectrometry (70,000 resolution at a mass/charge [$m/z$] ratio of 200), $^{15}N$ and $^{13}C$ fractions were separated on the basis of the mass defect induced by the neutron-binding energy. Neither cell growth in 2-D culture (Fig. 3b) nor the incorporation of [$^{13}C_5$/$^{15}N_2$]glutamine and subsequent production of labeled glutamate and aspartate in the $^{15}N$ and $^{13}C$ fractions (Fig. 3c–e and Supplementary Fig. 2a–e) differed substantially between TSM and AIG-3 cells. Incorporation of a stable isotope-labeled metabolite and its labeling efficiency at steady state serve as a basis for evaluation of the effects of changes in the expression of a metabolic enzyme on metabolic pathways[32]. Given that the labeling efficiency at steady state can reflect the activities of many metabolic reactions; however, it is difficult to evaluate effects on an individual reaction[33–35]. We therefore evaluated the labeling efficiency for each reaction at early (0.25 h) and late (6 and 24 h) time points after the onset of labeling[32]. Whereas a $^{13}C$ fraction of IMP, AMP, GMP, or UMP was not detected in TSM or AIG-3 cells, incorporation of $^{15}N$ into IMP (Fig. 3f), AMP (Fig. 3g), GMP (Fig. 3h), and UMP (Fig. 3i) of the de novo nucleotide biosynthesis pathway was increased markedly in AIG-3 cells at both 6 and 24 h after the onset of labeling compared with TSM cells. In AIG-3 cells, the labeling efficiency for nitrogen in IMP was increased not only in M + 1 and M + 2 fractions but also in M + 3 (Fig. 3f), suggesting that [$^{15}N$]aspartate originating from [$^{15}N$]glutamine also contributes to this labeling (Fig. 3e). Similar results were obtained for UMP (Fig. 3i). The increased efficiency for $^{15}N$-labeling of nucleotides (IMP, AMP, GMP, and UMP) in AIG-3 cells (Fig. 3f–i) was associated with a decrease in their pool size compared with that in TSM cells (Supplementary Fig. 2f–i). Given the marked upregulation of enzymes for nucleic acid synthesis and the marked increase in labeling efficiency of nucleotides in AIG-3 cells, this decrease in the pool size of nucleotide metabolites in these cells is likely attributable to an increased rate of nucleotide synthesis.

We next prepared culture medium containing amino acids at the physiological concentrations present in human plasma[36]. After culture in such physiological medium for 5 days, TSM and AIG-3 cells were labeled with 0.6 mM [$^{13}C_5$/$^{15}N_2$]glutamine. The

$^{15}N$- labeling efficiency for AMP and GMP was significantly greater in AIG-3 cells than in TSM cells (Supplementary Fig. 3), consistent with the original results obtained for the cells cultured in conventional medium (Fig. 3c–i). IMP was not detected under the physiological condition, likely because of the low concentration of glutamine.

To evaluate the activity of GLS1, we examined the isotopomer distribution for metabolites in the glutamine anaplerotic pathway into the TCA cycle at 0.25 h after the onset of labeling. Incorporation of $^{13}C$ from [$^{13}C_5$/$^{15}N_2$]glutamine into intermediates of the TCA cycle including α-KG (Fig. 3j and Supplementary Fig. 2j) and fumarate (Fig. 3k and Supplementary Fig. 2k) was reduced in AIG-3 cells compared with TSM cells. α-KG and fumarate are not only produced by the glutamine anaplerosis; they are also generated by the de novo nucleic acid synthesis pathway and are present in the cytosol (Supplementary Fig. 2l). Labeling of nucleic acid metabolites with $^{15}N$ was essentially not detected at 0.25 h; however, suggesting that the glutamine anaplerosis mediated by GLS1 was largely responsible for the production of α-KG and fumarate at this early time point. In contrast, the $^{13}C$-labeling efficiency for α-KG and fumarate at 6 or 24 h after the onset of labeling did not differ between the two cell types (Fig. 3j, k), suggesting that these metabolites at these time points reflect the sum of those derived from the GLS1- and PPAT-dependent pathways. The markedly reduced labeling efficiency for α-KG and fumarate at 0.25 h in AIG-3 cells compared with that in TSM cells (Fig. 3j, k), thus suggested that the activity of GLS1 was decreased in AIG-3 cells. Together, our proteomics and metabolomics data thus consistently indicated that the fate of glutamine is substantially shifted from the TCA cycle to the de novo nucleotide biosynthetic pathway during malignant transformation in an experimental cancer cell model.

**PPAT-GLS1 balance governs glutamine fate**. The expression of PPAT and GLS1 was increased and decreased, respectively, resulting in an increase in the PPAT/GLS1 ratio, during malignant transformation. To investigate whether the balance between these two enzymes is deterministic for cell proliferation, we examined the effects of GLS1 overexpression in AIG-3 cells (Fig. 4a and Supplementary Fig. 4a). A mutant (S286A) form of GLS1 that lacks enzymatic activity[37] was examined as a control to determine whether any effect of overexpression is dependent on GLS1 activity. Forced expression of GLS1, but not that of GLS1 (S286A), resulted in an increase in the labeling rate for α-KG as well as a decrease in that for IMP (Fig. 4b and Supplementary Fig. 4b), AMP (Supplementary Fig. 4c), GMP (Supplementary Fig. 4d), glutamate (Supplementary Fig. 4e), and aspartate (Supplementary Fig. 4f) in [$^{13}C_5$/$^{15}N_2$]glutamine metabolomics analysis. It also induced a decrease in the intracellular level of

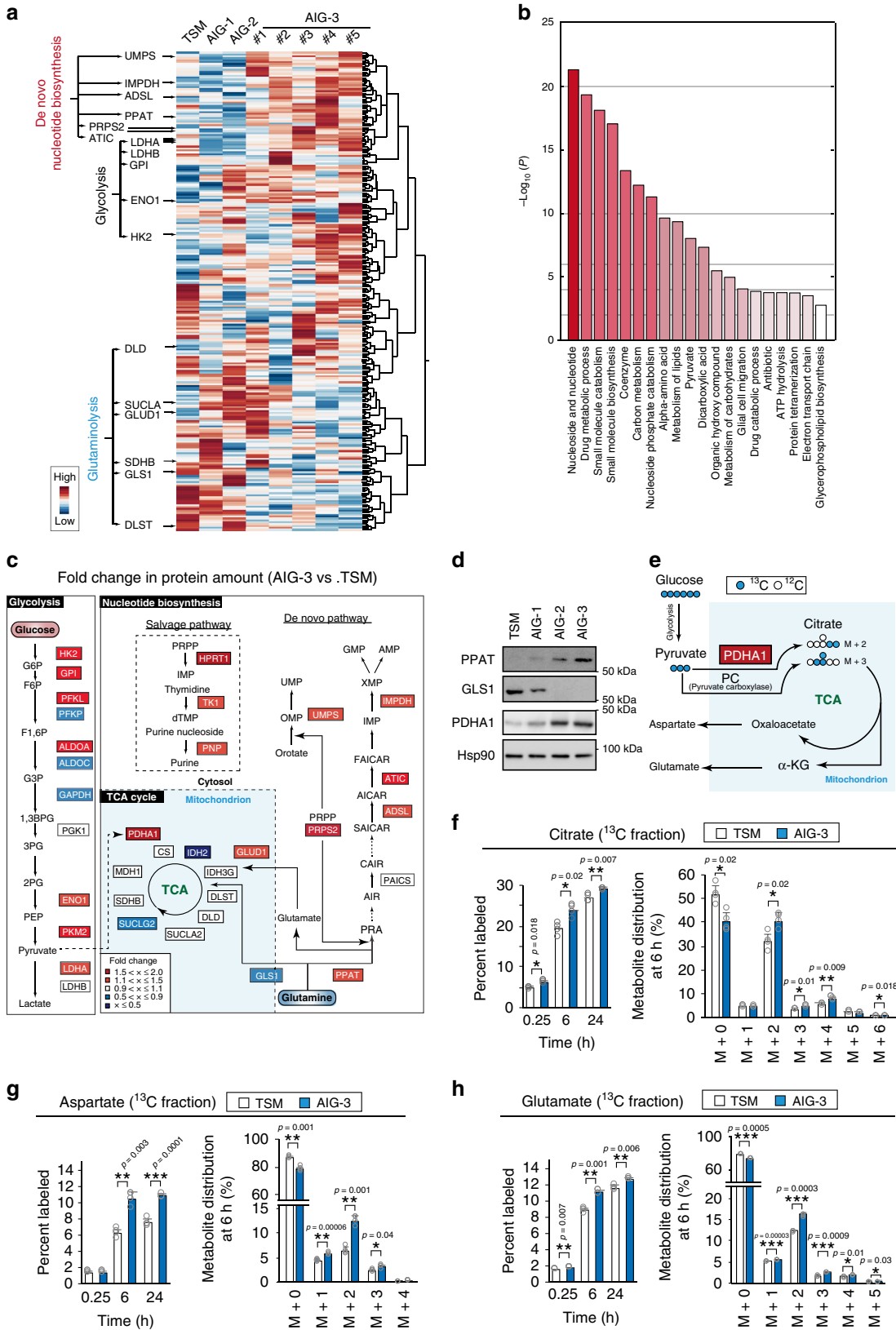

glutamine in AIG-3 cells (Supplementary Fig. 4g). The proliferation rate of AIG-3 cells in 2-D (Fig. 4c) or 3-D (Fig. 4d) culture was also significantly reduced by overexpression of GLS1. In addition, treatment with 100 nM CB-839, a GLS1 inhibitor, reversed the inhibitory effect of GLS1 overexpression on the proliferation of AIG-3 cells (Fig. 4c). AIG-3 cells overexpressing

GLS1 were not able to survive in glutamine-free medium, and the proliferation rate of these cells increased as the glutamine concentration of the medium was increased (Fig. 4e). Furthermore, tumors formed by the GLS1-overexpressing cells in nude mice were significantly smaller than those formed by control cells (Fig. 4f). The metabolic disturbance induced by excessive activity

**Fig. 2 Global view of proteomic changes in malignant cancer. a** Heat map for hierarchical clustering of metabolic enzymes measured by iMPAQT in TSM and AIG cell lines. The abundance of enzymes of the de novo nucleotide biosynthetic pathway was increased, whereas that of those of the glutaminolysis pathway was reduced, in AIG-3 cell lines compared with TSM cells. Measurements were conducted with three biological replicates ($n = 3$). Five independent AIG-3 lines (#1 to #5) were examined. Scale bar indicates z-scores for protein amount. **b** GO analysis of metabolic enzymes whose abundance as revealed by iMPAQT was changed in AIG-3 cells compared with TSM cells. The analysis was conducted with Metascape (two-sided, $P \leq$ 0.05), and adjusted $P$-values are shown (Benjamini-Hochberg false discovery rate). **c** iMPAQT-determined changes in the abundance of enzymes in the indicated metabolic pathways for AIG-3 versus TSM cells. **d** Immunoblot analysis of PPAT, GLS1, and PDHA1 in TSM and AIG-1 to -3 cells, was conducted with single time. **e** Schematic representation of [$^{13}C_6$]glucose metabolism. Blue and white circles represent $^{13}C$ and $^{12}C$, respectively. **f–h** TSM and AIG-3 cells were exposed to 17.5 mM [$^{13}C_6$]glucose in monolayer culture for up to 24 h, after which the percentage of and distribution at 6 h for $^{13}C$-labeled citrate, aspartate, and glutamate, respectively, were measured by ion chromatography and mass spectrometry (IC-MS or LC-M). Metabolite measurements were conducted with four biological replicates, and data are means + s.d. *$P < 0.05$, **$P < 0.01$, ***$P < 0.001$ (paired two-tailed Student's t-test). Source data are provided as a Source Data file.

of GLS1 may thus have reduced glutamine availability for nucleic acid synthesis and thereby inhibited cell proliferation in the GLS1-overexpressing AIG-3 cells. Collectively, these results also suggested that excessive activity of GLS1 inhibits tumor growth.

In a converse approach, we examined the effects of PPAT depletion by short hairpin RNA (shRNA)-mediated RNA interference in AIG-3 cells (Fig. 4g and Supplementary Fig. 4a). PPAT depletion markedly inhibited labeling of IMP, AMP, and GMP with $^{15}N$ derived from [$^{13}C_5/^{15}N_2$]glutamine (Fig. 4h and Supplementary Fig. 5a–c), whereas it increased the incorporation of $^{13}C$ into α-KG (Fig. 4h and Supplementary Fig. 5a). The labeling efficiency, pool size, and isotopologue distribution of glutamine (Supplementary Fig. 5d), glutamate (Supplementary Fig. 5e), and aspartate (Supplementary Fig. 5f) did not differ significantly between control and PPAT-depleted AIG-3 cells. The proliferation rate of AIG-3 cells in 2-D (Fig. 4i) or 3-D (Fig. 4j) culture was also reduced by depletion of PPAT. Given that many nucleotides including AMP and GMP exert feedback inhibition on PPAT and other enzymes in the de novo nucleotide biosynthesis pathway, the supplementation of AMP or GMP to PPAT-depleted AIG-3 cells did not improve their capacity for growth (Supplementary Fig. 6a). Overexpression of human PPAT, but not supplementation with hypoxanthine to activate the salvage pathway of nucleotide biosynthesis, normalized the anchorage-independent growth of PPAT-depleted AIG-3 cells (Supplementary Fig. 6b). Furthermore, the PPAT-depleted cells formed smaller tumors in nude mice than did control cells (Fig. 4k). However, depletion of GLS1 or overexpression of PPAT alone in TSM cells did not confer a malignant phenotype similar to that of AIG-3 cells (Supplementary Fig. 6c, d), suggesting that downregulation of GLS1 or upregulation of PPAT may be required but not sufficient for malignant transformation. Exposure to dimethyl α-KG, a membrane-permeable form of α-KG, did not affect the proliferation of AIG-3 cells (Supplementary Fig. 6e), excluding the possibility that changes in gene expression induced by an altered level of α-KG account for observed changes in cell proliferation. Together, these various data suggested that a reduced activity of PPAT inhibits tumor growth, and they were consistent with the notion that the balance between GLS1 and PPAT governs the metabolism of carbon and nitrogen derived from glutamine and thereby controls cell proliferation and tumor growth.

**Effects of GLS inhibition depend on glutamine concentration.** We next examined the effects of glutamine starvation and the GLS1 inhibitor CB-839[14,38–40] on TSM and AIG-3 cells maintained in conventional culture medium (2 mM glutamine) or in the physiological medium (0.6 mM glutamine) mentioned above[36]. The growth-inhibitory effects of glutamine deprivation or CB-839 treatment were less pronounced for TSM or AIG-3 cells maintained in the physiological medium than for those

maintained in conventional medium (Supplementary Fig. 7a–d). In addition, AIG-3 cells, in which the activity of GLS1 is attenuated compared with that in TSM cells (Fig. 3c–k), tended to show more resistance to glutamine starvation and CB-839 treatment compared with TSM cells (Supplementary Fig. 7a–d). The sensitivity to glutamine deprivation was thus similar to that to CB-839 treatment, consistent with previous observations showing that the sensitivity to glutamine starvation depends on the activity of the rate-limiting enzyme GLS1[14,38–40]. The extracellular glutamine concentration has been found to affect the expression of metabolic enzymes[41]. We therefore applied immunoblot analysis to measure the abundance of GLS1 and PPAT in TSM and AIG-3 cells cultured in the physiological or conventional medium. We found that the PPAT/GLS1 ratio was greatly increased in TSM or AIG-3 cells exposed to the physiological medium compared with the corresponding cells maintained in conventional medium (Supplementary Fig. 7e, f). Collectively, these results thus suggested that the presence of excess glutamine increases the dependence of cancer cells on GLS1, with the possible consequence that CB-839 inhibits the growth of such cells selectively and is less effective for cancer cells under the physiological condition.

**Perturbation of PPAT-GLS1 balance in human cancer cell lines.** To investigate the generalizability of the notion that the balance between PPAT and GLS1 is deterministic for proliferation of human cancer cell lines, we examined the effects of overexpression of these enzymes in A549 and HeLa cells (Supplementary Fig. 8a), which preferentially rely on pyruvate for TCA anaplerosis[15,16]. Forced expression of GLS1 reduced the proliferation rate of A549 and HeLa cells in 2-D (Supplementary Fig. 8b) or 3-D (Supplementary Fig. 8c) culture as well as the tumorigenicity of these cells in nude mice (Supplementary Fig. 8d). On the other hand, overexpression of PPAT promoted the proliferation of both cell lines in 2-D and 3-D culture as well as their tumorigenicity in nude mice, although, with the exception of that on anchorage-independent growth of HeLa cells, these effects did not achieve statistical significance (Supplementary Fig. 8b–d). We also examined the effects of GLS1 or PPAT depletion in A549 and HeLa cells (Supplementary Fig. 8e). GLS1 depletion in A549 and HeLa cells did not affect proliferation in 2-D culture (Supplementary Fig. 8f) or tumor formation in nude mice (Supplementary Fig. 8h), whereas it slightly increased anchorage-independent growth (Supplementary Fig. 8g). In contrast, depletion of PPAT in both cell lines inhibited proliferation in 2-D (Supplementary Fig. 8f) and 3-D (Supplementary Fig. 8g) culture as well as tumor formation in nude mice (Supplementary Fig. 8h). Overall, the effects of modulation of PPAT-GLS1 expression balance in cancer cell lines on cell proliferation and tumorigenesis in nude mice were consistent with those observed in transformed fibroblasts.

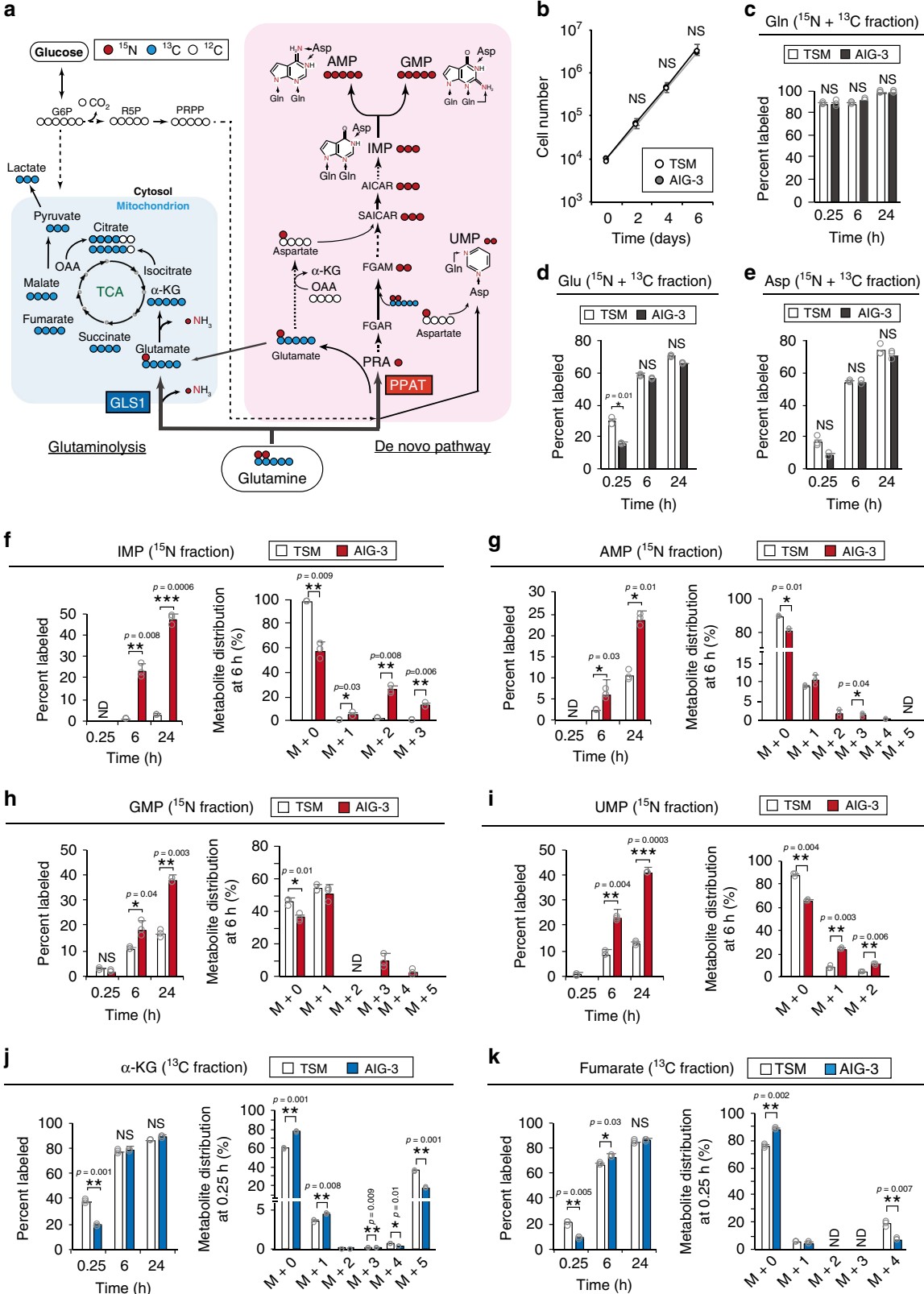

**PPAT-GLS1 balance as a potential therapeutic target**. Recent studies have suggested that dependence on glutamine metabolism might differ among cancer types[18]. We therefore next applied immunoblot analysis to measure the abundance of metabolic enzymes including GLS1 and PPAT in 118 cancer cell lines originating from a variety of human tissues. We found that the PPAT/GLS1 ratio varied substantially among these cell types (Fig. 5a). Breast, brain, and lung cancer cells manifested a relatively high PPAT/GLS1 ratio, whereas stomach and colorectal cancer cells showed a low PPAT/GLS1 ratio. Cells with a high PPAT/GLS1 ratio such as BT-549 breast cancer cells (Fig. 5b), TGW neuroblastoma cells (Fig. 5c), and U-251 MG glioblastoma

**Fig. 3 The balance of glutamine utilization is skewed away from glutaminolysis toward nucleotide biosynthesis during malignant progression. a** Schematic representation of $[^{13}C_5/^{15}N_2]$glutamine metabolism. $[^{13}C_5/^{15}N_2]$Glutamine is anabolized to generate nucleotides via the de novo nucleotide biosynthetic pathway, whereas it is catabolized via the glutaminolysis pathway to support anaplerotic reactions for the TCA cycle. Red circles represent $^{15}N$, blue circles $^{13}C$, and white circles $^{12}C$. **b** Proliferation rate of TSM and AIG-3 cells in 2-D culture with medium containing 2 mM glutamine ($n = 3$). **c–e** TSM and AIG-3 cells were exposed to 2 mM $[^{13}C_5/^{15}N_2]$glutamine in monolayer culture for up to 24 h, after which the percentage of $^{13}C_5/^{15}N_2$-labeled glutamine, $^{13}C_5/^{15}N_1$-labeled glutamate, and $^{15}N_1$-labeled aspartate was measured by IC-MS. **f–i** TSM and AIG-3 cells were exposed to 2 mM $[^{13}C_5/^{15}N_2]$ glutamine in monolayer culture for up to 24 h, after which the percentage of and label distribution at 6 h for $^{15}N$-labeled IMP (**f**), AMP (**g**), GMP (**h**), or UMP (**i**) were measured by IC-MS or LC-M. **j, k** TSM and AIG-3 cells were exposed to 2 mM $[^{13}C_5/^{15}N_2]$glutamine in monolayer culture for up to 24 h, after which the percentage of and distribution at 15 min for $^{13}C$-labeled metabolites of glutaminolysis (α-KG and fumarate, respectively) were measured by IC-MS or LC-M. Measurements in **c–k** were performed with an Orbitrap-type mass spectrometer (70,000 resolution at $m/z$ 200), and $^{15}N$ and $^{13}C$ fractions were separated on the basis of the mass defect induced by the neutron-binding energy. The proportion of $^{15}N$ and $^{13}C$, of $^{15}N$, or of $^{13}C$ in each metabolite was calculated from the mass isotopomer distribution determined by IC-MS or LC-M S. All metabolite measurements were conducted with three biological replicates for each experiment, and all data are means ± s.d. ND, not detected. *$P < 0.05$, **$P < 0.01$, ***$P < 0.001$; NS, not significant (paired two-tailed Student's $t$-test). Source data are provided as a Source Data file.

cells (Fig. 5d) were sensitive to PPAT depletion or GLS1 over-expression (Supplementary Fig. 9) with regard to their anchorage-independent growth. In contrast, GLS1 depletion or PPAT overexpression did not consistently increase the anchorage-independent growth of these cells. Conversely, for cells with a low PPAT/GLS1 ratio such as HCT116 colorectal cancer cells (Fig. 5e) and HSC-60 gastric cancer cells (Fig. 5f), PPAT depletion or GLS1 overexpression (Supplementary Fig. 9) promoted (HCT116) or had no effect (HSC-60) on anchorage-independent growth, whereas such growth of both cell lines was substantially attenuated by GLS1 depletion or PPAT overexpression. We next examined whether these various cancer cell lines cultured in the physiological medium were susceptible to CB-839. The growth-inhibitory effect of CB-839 treatment apparent for some of the cell lines in conventional medium (2 mM glutamine) was lost or substantially attenuated in the physiological medium (0.6 mM glutamine) (Fig. 5g–n). Indeed, only HCT116 colorectal cancer cells (Fig. 5m) and HSC-60 gastric cancer cells (Fig. 5n) showed a significant growth-inhibitory effect of CB-839 under the physiological condition. In addition, the PPAT/GLS1 ratio for these various cell lines was greater under the physiological condition than in the presence of excess glutamine (Supplementary Fig. 10). These results were consistent with the effects of perturbation of PPAT and GLS1 expression levels (Fig. 5b–f), and they suggested that cancers of gastrointestinal origin are substantially dependent on glutamine anaplerosis, similar to normal gastrointestinal epithelial cells (see Discussion).

Collectively, our data indicated that modulation of glutamine fate might be of therapeutic value for many types of cancer, with the potential antitumor effect possibly being predictable on the basis of the endogenous expression levels of GLS1 and PPAT.

**PPAT is strongly associated with cancer prognosis.** To validate our findings that PPAT-GLS1 balance influences cancer malignancy in cells and in mice, we examined the relation of PPAT and GLS1 gene expression to overall survival in single cohorts of lung, brain, or neuroendocrine cancer patients (Supplementary Fig. 11a). The outcome of individuals with high PPAT expression was significantly worse than that of those with low PPAT expression in each cohort, whereas individuals with low GLS1 expression tended to show a poorer prognosis compared with those with high GLS1 expression. These differences were most pronounced in the patients with neuroendocrine cancer.

We next extended these findings by performing an integrated meta-analysis for all metabolic enzymes with public datasets representing a total of ~11,000 patients with a variety of cancers (Fig. 6, and Supplementary Figs. 11b–d and 12). Many enzymes of the de novo purine biosynthesis pathway were found to have a high hazard ratio (HR) with regard to their gene expression level

and overall survival in these combined cancer cohorts, with three of them—PPAT, glycinamide ribonucleotide transformylase (GART), and GMP synthase (GMPS)—being ranked in the top four (Fig. 6a). In contrast, enzymes of the glutamine anaplerotic pathway into the TCA cycle—such as glutamate dehydrogenase 1 (GLUD1) and GLS2—did not have a high HR or tended to be associated with good prognosis. The HR for PPAT was remarkably high in tissue-specific cohorts, such as those for lung, breast, brain, hematopoietic, neuroendocrine, liver, or pancreatic cancer, in both the random-effects model and the fixed-effects model (Fig. 6b and Supplementary Fig. 11b), whereas the expression of GLS1 was not significantly related to cancer prognosis in any cancer type with the exception of colorectal cancer. Of note, PPAT and GLS1 expression was significantly associated with poor (HR = 5.21) or tended to be associated with good (HR = 0.38) prognosis, respectively, in neuroendocrine cancer (Fig. 6b). The relation of the PPAT/GLS1 ratio to prognosis in colorectal cancer appeared opposite to that in the other types of cancer. Instead of PPAT, hypoxanthine phosphoribosyltransferase 1 (HPRT1) was significantly associated with poor prognosis in colorectal cancer (Supplementary Fig. 10c, d), suggesting that the salvage pathway of nucleotide biosynthesis is dominant over the de novo pathway in this cancer type. Meta-analysis also revealed that prognostic prediction based on the PPAT/GLS1 ratio reproduced that based on PPAT and GLS1 separately (Supplementary Fig. 13). However, the effect of the PPAT/GLS1 ratio was slightly smaller than that of PPAT alone, suggesting that the positive effect of PPAT and the negative effect of GLS1 on the HR do not correspond in a simple one-to-one manner. Together with our observations in cells and mice showing that PPAT-GLS1 balance regulates cell proliferation and malignancy, these results with human cohorts thus suggested that the expression of PPAT is a strong indicator for prognosis in many cancer types, consistent with the findings of a previous study of a single lung cancer cohort[21].

**PPAT as a new therapeutic target for SCLC.** The observation that PPAT expression was significantly associated with poor prognosis for neuroendocrine cancer prompted us to investigate whether such expression is also related to the outcome of SCLC, which is thought to be a high-grade neuroendocrine cancer[23]. Indeed, RNA-sequencing data for SCLC patients (GSE60052) revealed that PPAT expression was greater in tumor than in normal tissue, whereas GLS1 expression was lower in tumor than in normal tissue (Fig. 7a). Kaplan-Meier analysis of the RNA-sequencing data for these SCLC patients[42] showed that the outcome of individuals with high PPAT expression in tumor tissue was significantly worse than that of those with low such expression (Fig. 7b). A similar pattern was observed for expression of

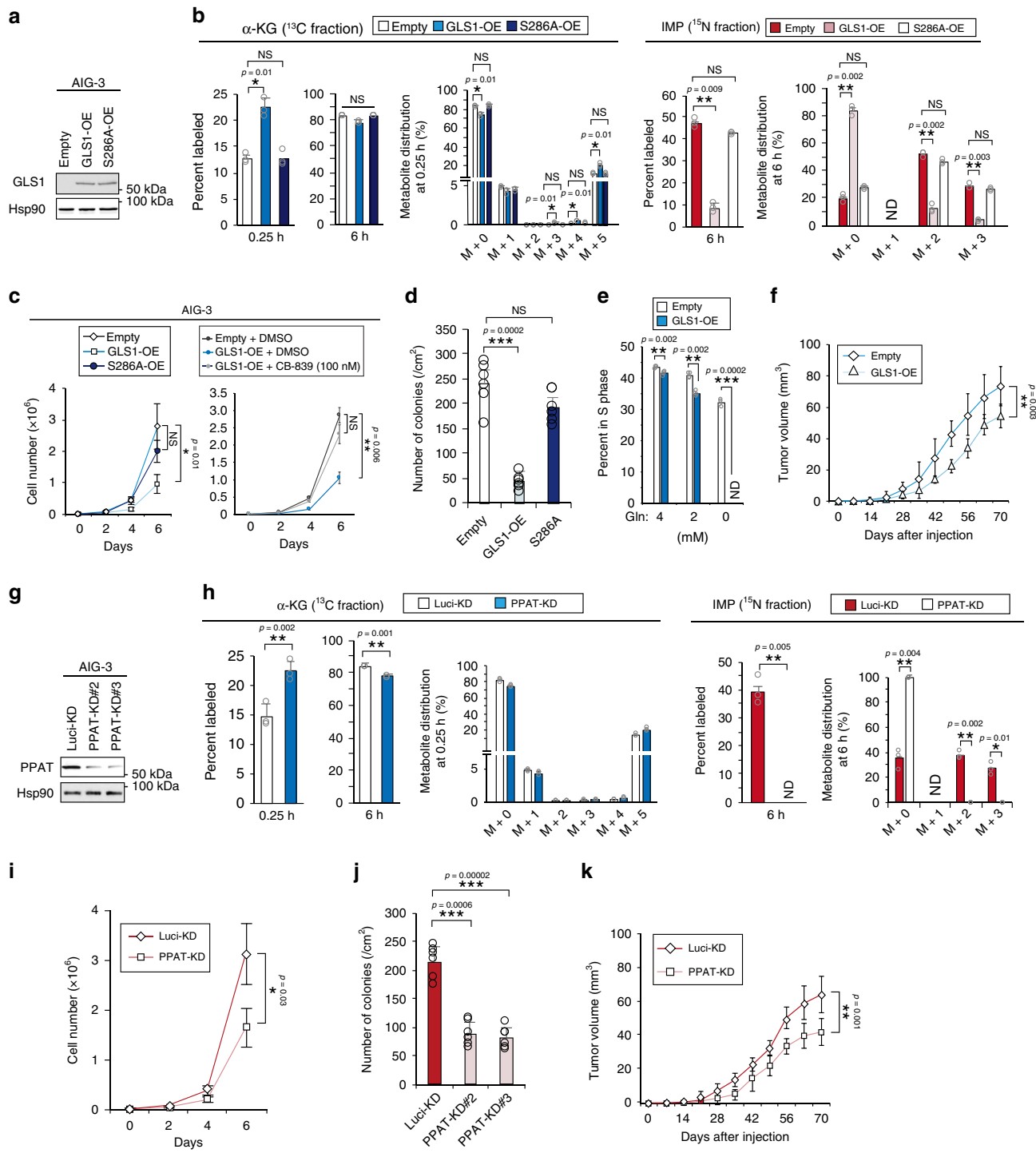

phosphoribosyl pyrophosphate synthetase 2 (PRPS2), which is also a rate-limiting enzyme in the de novo nucleotide synthesis pathway whose expression is regulated by c-Myc[19,20], although the difference in overall survival was smaller than that for PPAT (Fig. 7c). In contrast, the expression of neither UMP synthetase (UMPS) (Fig. 7d), a rate-limiting enzyme for pyrimidine synthesis, nor GLS1 (Fig. 7e) appeared to be related to the prognosis of SCLC patients.

We next depleted three SCLC cell lines (MS-1-L, STC-1, and SBC-3) of PPAT with specific shRNAs and examined the effects on cell growth in 3-D culture. Depletion of PPAT inhibited the anchorage-independent growth of all three cell lines (Fig. 7f–k), and this effect in SBC-3 cells was reversed by forced expression of

PPAT in a manner resistant to its shRNA-mediated knockdown (Fig. 7j, k). Overexpression of GLS1 also suppressed the anchorage-independent growth of SBC-3 cells (Fig. 7l, m). Collectively, the strong correlation of PPAT expression with poor prognosis in SCLC patients as well as the suppression of anchorage-independent growth of SCLC cell lines by PPAT depletion indicated that PPAT might be a promising therapeutic target for SCLC.

## Discussion

In addition to the carbon shift of the Warburg effect, here we report that a nitrogen shift plays a key role in malignant progression of cancer. Our results show that both PPAT and GLS1

**Fig. 4 PPAT-GLS1 balance governs glutamine fate. a** Immunoblot analysis of GLS1 in AIG-3 cells stably overexpressing (OE) wild-type or S286A mutant forms of human GLS1 or infected with the corresponding empty retrovirus, was conducted with single time. **b** Cells as in **a** were exposed to 2 mM [$^{13}C_5$/$^{15}N_2$]glutamine in monolayer culture for 0.25 or 6 h, after which the percentage and distribution of $^{13}C$-labeled α-KG or $^{15}N$-labeled IMP were measured by IC-MS analysis. ($n = 3$). ND, not detected. $^{15}N$-Labeled IMP was not detected at 0.25 h. **c** Proliferation rate of cells as in **a** in 2-D culture with 2 mM glutamine and in the absence (left panel) or presence (right panel) of 100 nM CB-839 or dimethyl sulfoxide (DMSO) vehicle ($n = 3$). **d** Anchorage-independent growth for cells as in **a** ($n = 6$). **e** AIG-3 cells stably overexpressing wild-type GLS1 or infected with the corresponding empty retrovirus were cultured in medium containing 0, 2, or 4 mM glutamine, and the proportion of cells in S phase was determined by flow cytometric analysis of 5-bromo-2-deoxyuridine incorporation ($n = 3$). **f** Tumorigenicity in nude mice for cells as in **e** ($n = 6$). **g** Immunoblot analysis of PPAT in AIG-3 cells stably infected with retroviruses encoding luciferase control (Luci-KD) or two independent PPAT shRNAs, was conducted with single time. **h** Cells as in **g** were exposed to 2 mM [$^{13}C_5$/$^{15}N_2$]glutamine in monolayer culture for 0.25 or 6 h, after which the percentage and distribution of $^{13}C$-labeled α-KG or $^{15}N$-labeled IMP were measured by IC-MS analysis ($n = 3$). $^{15}N$-Labeled IMP was not detected at 0.25 h. **i–k** Proliferation rate in 2-D culture with 2 mM glutamine ($n = 3$) (**i**), anchorage-independent growth ($n = 6$) (**j**), and tumorigenicity in nude mice ($n = 6$) (**k**) for cells as in **g**. All cells count and metabolite measurements were conducted with three biological replicates. All colonies were counted in two randomly selected fields in each of three dishes. All quantitative data are means ± s.d. *$P < 0.05$, **$P < 0.01$, ***$P < 0.001$ (paired two-tailed Student's $t$-test). Source data are provided as a Source Data file.

are decisive factors in control of the shift in utilization of nitrogen derived from glutamine, which affects the efficiency of malignant transformation both in vitro and in vivo. Our experimental findings with cells and mice together with statistical data from an integrated meta-analysis in human cohorts support the notion that metabolic convergence of glutamine toward nucleotide biosynthesis may be an almost universal process in malignant progression of human cancer (Fig. 8).

Although GLS1 has been thought to have a protumor effect, we now show that it can actually and partially have an antitumor effect, as revealed by our experimental results for metabolomics analysis, anchorage-independent growth, and tumor formation in nude mice as well as our meta-analysis of human cohorts. Glutamine oxidation was previously found to be suppressed in anchorage-independent spheroids compared with monolayer-cultured cells[43]. Furthermore, neither genetic ablation nor pharmacological inhibition of GLS1 was found to affect the growth of lung tumors in vivo[16]. GLS2, an isozyme of GLS1, was also shown to have antitumor potential[44,45]. Given that the core region of solid tumors is glutamine deficient[46], nitrogen metabolism in cells in this region is likely directed toward nucleotide biosynthesis through an increase in PPAT expression and suppression of GLS1 so as to support cell proliferation under the condition of glutamine limitation. Under this condition, carbon derived from glucose, rather than from glutamine, contributes to anaplerosis for the TCA cycle[26,47,48] in association with an increase in the abundance of PDHA1[16]. Although catabolism of glutamine via GLS1 might be required for ATP synthesis in cancer, little evidence supports the notion that excessive accumulation of ATP in cells promotes cell proliferation[18,49,50].

Our conclusion that GLS1 is not a protumor factor in many cancer types (with the exception of colorectal cancer) is supported by the results of our meta-analysis, an approach that integrates data from many independent cohorts and provides the highest level of evidence[51,52]. Whereas most types of cancer show a high HR for PPAT, colorectal cancer is an exception, with HRs for PPAT and GLS1 being 0.96 and 1.12, respectively. The high dependence of colorectal cancer on GLS1 might reflect an original trait of intestinal epithelial cells, which utilize glutamine as a metabolic fuel instead of glucose[53–56]. Promotion of glutamine anaplerotic pathway into the TCA cycle in the intestinal epithelium is thought to reduce the consumption of glucose by this tissue and thereby allow its efficient transport to the bloodstream. Given that growth and disease progression are relatively slow in colorectal cancer compared with other cancer types[57–59].

In conclusion, we have identified key factors that control the metabolic fate of glutamine and the dependence of tumors of different organs on these factors. Our findings provide the basis for exploration of a different regime for cancer treatment. Among

the identified factors, PPAT may be one of the most promising targets, given its substantial contribution to the nitrogen shift from glutamine as well as its high association with prognosis in many cancer types.

## Methods

**Cell culture and oncogene-induced transformation.** TIG-3 human embryonic lung diploid fibroblasts (Japanese Collection of Research Bioresources) were cultured under an atmosphere of 5% $CO_2$ at 37 °C in Dulbecco's modified Eagle's medium (DMEM, Wako Pure Chemical) supplemented with 10% fetal bovine serum (FBS, Life Technologies), 1 mM pyruvate (Gibco), and antibiotics. Human cancer cell lines (Laboratory of Proteome Research, National Institute of Biomedical Innovation, Health, and Nutrition) were cultured under an atmosphere of 5% $CO_2$ at 37 °C in RPMI medium supplemented with 10% FBS, 1 mM pyruvate, and antibiotics. We confirmed that TIG-3 cells and human cancer cell lines were free of mycoplasma contamination. For retroviral infection, the mouse ecotropic retrovirus receptor was introduced into cells with the use of an amphotropic virus produced by HEK293T cells transfected with both pCX4-EcoVR and pGP/pE-ampho (Takara Bio). Complementary DNAs for hTERT, the SV40 early region, and human c-Myc in pCX4 were introduced into HEK293T cells together with pGP/pE-eco (Takara Bio) for the production of recombinant retroviruses. TIG-3 cells expressing the mouse ecotropic retrovirus receptor were then infected with the retrovirus encoding hTERT. The resulting cells, designated TIG-3(T), were further infected with the retrovirus for the SV40 early region and subjected to selection to yield TIG-3(TS) cells, which were then infected with the retrovirus for c-Myc to establish TIG-3(TSM) cells.

**Glucose uptake.** Cells were cultured for 30 min in medium containing 100 µM 2-[N-(7-nitrobenz-2-oxa-1,3-diazol-4-yl)amino]-2-deoxy-D-glucose (2-NBDG; Peptide Institute, 23002-v), washed once with phosphate-buffered saline (PBS), resuspended in PBS containing propidium iodide (5 µg/ml), and analyzed by flow cytometry.

**RT and real-time PCR analysis.** Total RNA isolated from cells with the use of the TRIzol reagent (Thermo Fisher Scientific) was subjected to reverse transcription (RT) with the use of ReverTra Ace (Toyobo), and the resulting cDNA was subjected to real-time polymerase chain reaction (PCR) analysis with SYBR Green PCR Master Mix and specific primers in a Step One Plus Real-Time PCR System (Applied Biosystems). Total microRNA was isolated from cells with the use of a NucleoSpin device (Takara Bio). The sequences of the various primers (sense and antisense, respectively) were as follows: 5′-TGCATGTTCCTGGCCTCC-3′ and 5′-TTAAAGTCGGTGGCACACA-3′ for CCND2, 5′-TTCAAATCACACAAGGG AATG-3′ and 5′-AACGAAGGGCTGACAATTTTCTA-3′ for PPAT, 5′-CGAAG AGAAGGTGGTGATCAAAG-3′ and 5′-GCAGGCTTCCAGCAAAAATTTAA-3′ for GLS1, 5′-GAAGACTCTGCACCCAGATTTA-3′ and 5′-TCTCTGCCAA ATCTGCTACAGAG-3′ for LDHA, 5′-GGAACAGAAAGACTATGACTCGCG-3′ and 5′-CTGCTGCCTCAGCCTGAGTT-3′ for GLS2, and 5′-TTGCCGA-CAGGATGCAGAAGGA-3′ and 5′-AGGTGGACAGCGAGGCCAGGAT-3′ for β-actin. The sense primer for mir-23a was 5′-ATCACATTGCCAGGGATTTCC-3′, and the antisense primer for mir-23a and primers for U6 were obtained from Mir-X kits (Clontech). The abundance of target mRNAs was normalized by that of β-actin mRNA, whereas that of mir-23a was normalized by that of U6 RNA.

**RNA interference.** For the establishment of cell lines expressing shRNAs for PPAT or GLS1, we designed and constructed shRNA vectors in pCX4. The target sequences were 5′-TCCCTGTCTAACTGTAGACAAA-3′ and 5′-CAGG GAAAGTACTAAACCAAAA-3′ for PPAT shRNAs #2 and #3, respectively;

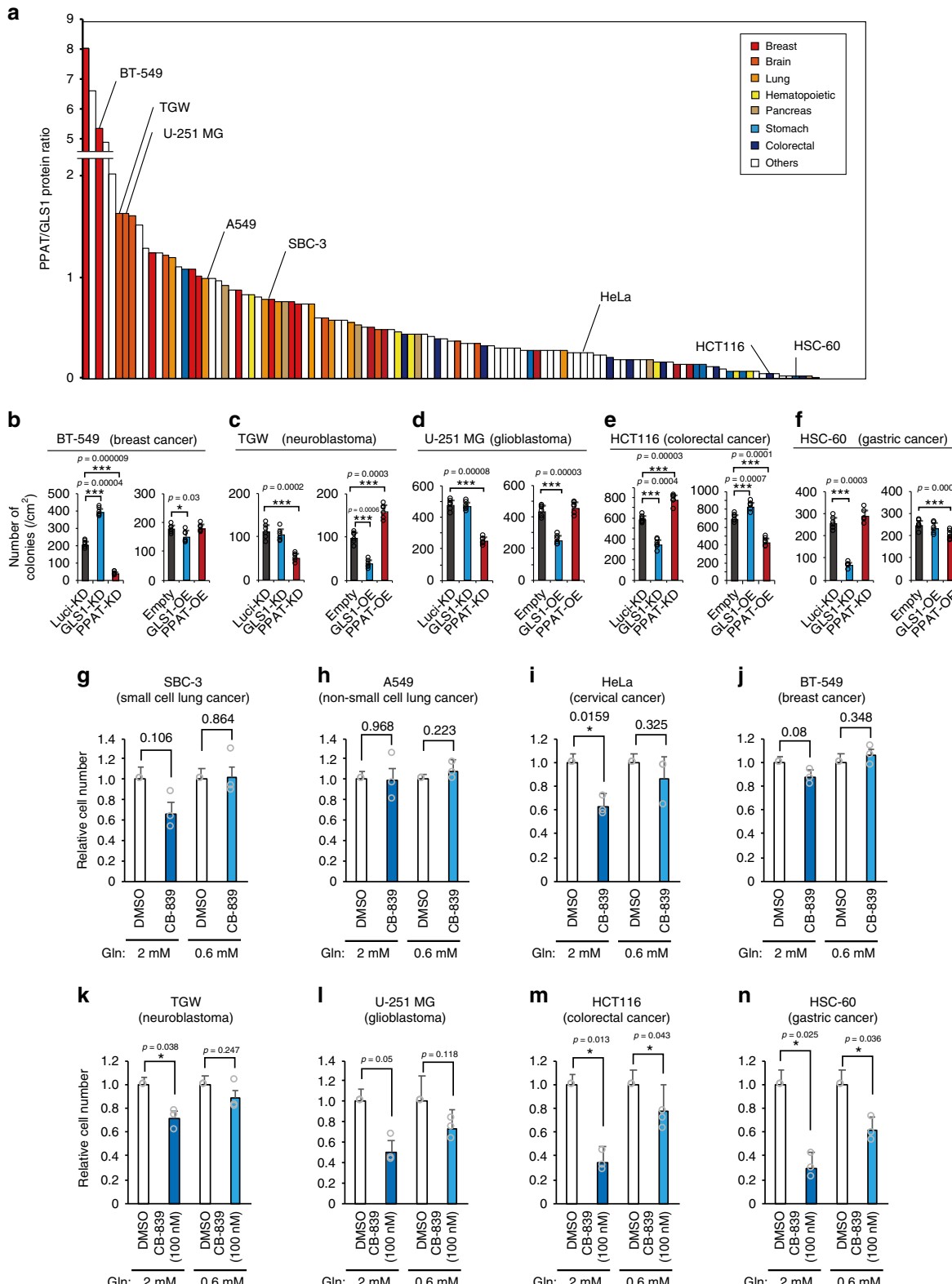

**Fig. 5 Perturbation of the PPAT/GLS1 ratio in human cancer cell lines. a** The PPAT/GLS1 ratio in 118 human cancer cell lines as determined by immunoblot analysis and densitometry. **b–f** Effects of knockdown (KD) or overexpression (OE) of GLS1 or PPAT on anchorage-independent growth were determined in the indicated cell lines ($n = 6$). **g–n** Equal numbers of cells ($5 \times 10^5$) of the indicated lines were plated and then cultured in medium containing 2 or 0.6 mM glutamine and in the presence of vehicle alone (DMSO) or 100 nM CB-839. Cell number was determined at 48 h and expressed relative to the corresponding value for the condition of 2 mM glutamine and DMSO ($n = 3$). All colonies were counted in two randomly selected fields in each of three dishes. All cells were counted with three biological replicates. Data in **b–n** are means + s.d. *$P < 0.05$, **$P < 0.01$, ***$P < 0.001$ (paired two-tailed Student's $t$-test). Source data are provided as a Source Data file.

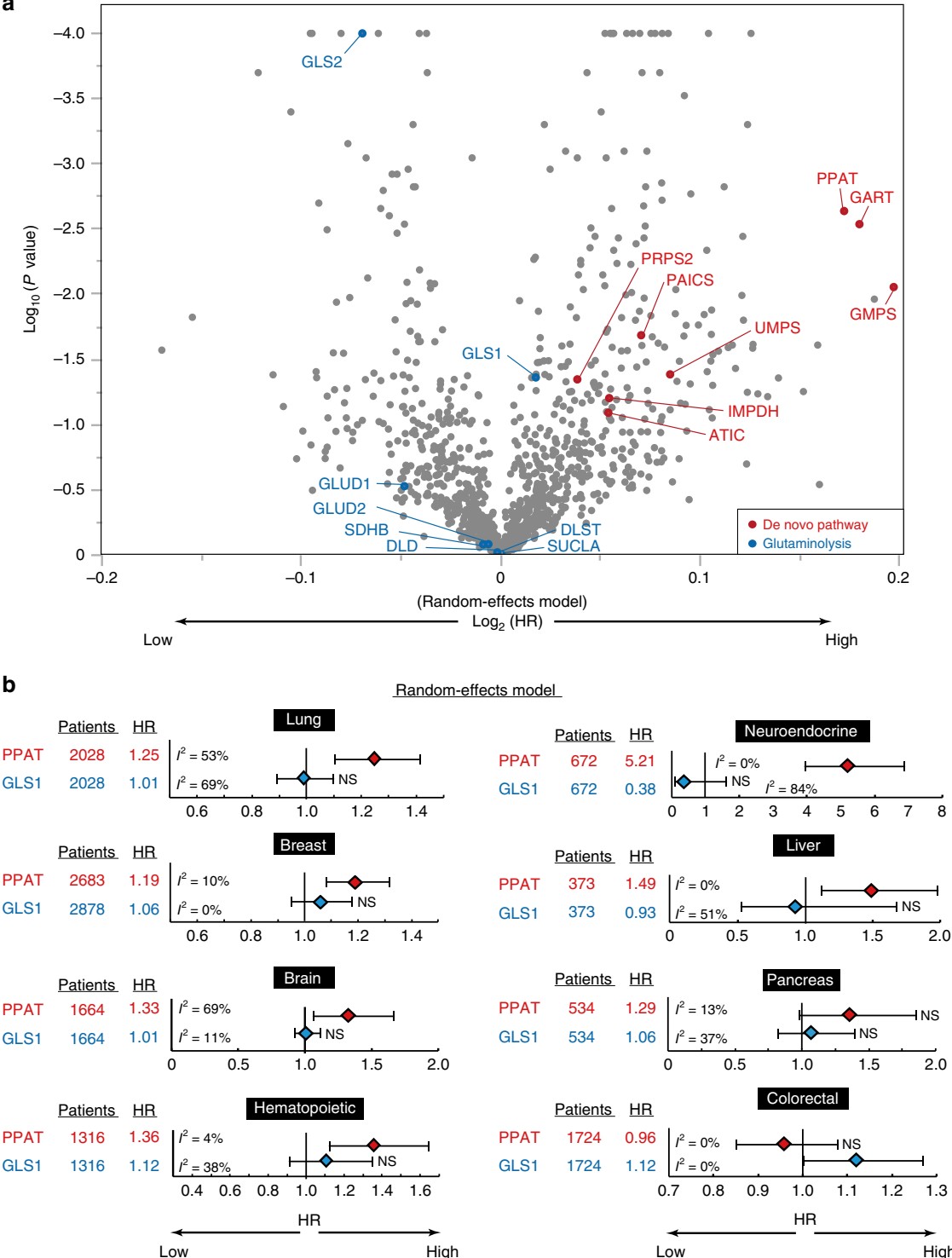

**Fig. 6 The PPAT/GLS1 ratio is strongly associated with cancer prognosis. a** Integrated meta-analysis for gene expression levels of all metabolic enzymes (1126 enzymes) in cancer cohort studies combined by means of the random-effects model. Enzymes of the de novo nucleotide biosynthetic pathway and the glutaminolysis pathway are shown in red and blue, respectively. The integrated hazard ratios (HRs) and $P$-values are shown in a volcano plot. The total numbers of patients were as follows: PPAT, 10,994; PRPS2, 11,401; GART, 12,135; PAICS, 11,286; ATIC, 11,146; IMPDH, 11,345; GMPS, 11,705; UMPS, 11,543; GLS1, 11,819; GLS2, 10,338; GLUD1, 10,898; GLUD2, 10,735; DLST, 11,217; DLD, 11,974; SUCLA, 10,865; and SDHB, 11,694. **b** Cancer cohort studies for each organ were combined by means of the random-effects model. The integrated HR and its 95% confidence interval, the numbers of patients, and the heterogeneity score ($I^2$) are shown. The centre of effect sizes (HR = 1.0) are shown as vertical line in **b**. NS not significant. All cohorts were divided at the median gene expression level in both **a**, **b**. Source data are provided as a Source Data file.

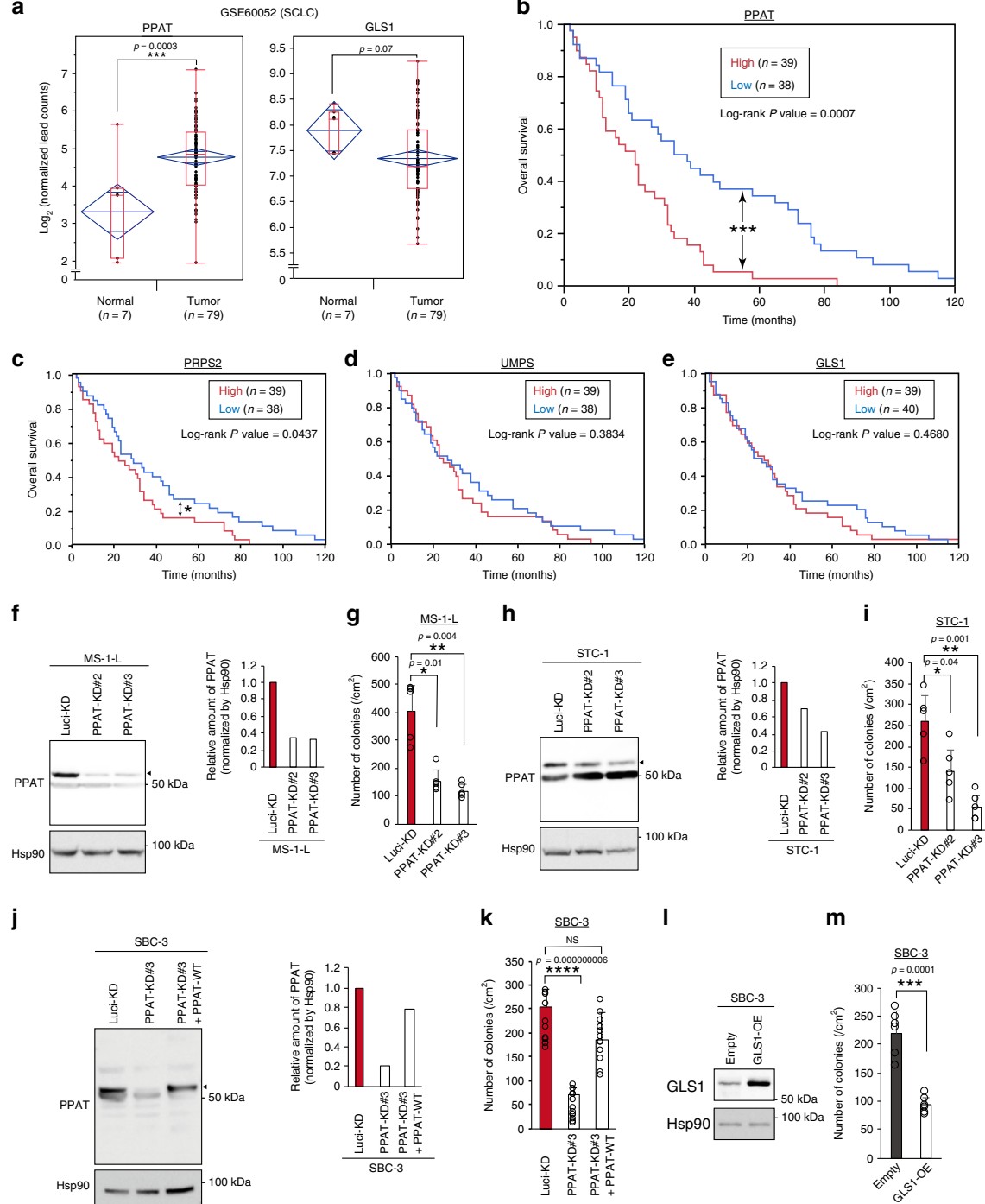

**Fig. 7 PPAT as a new therapeutic target for SCLC. a** Box plots of PPAT and GLS1 gene expression levels in normal lung tissue ($n = 7$) or tumors ($n = 79$) from SCLC patients (GSE60052 data set). The upper and lower limits of the red boxes represent quartiles, with the line within the boxes indicating the median and the whiskers showing the extremes. Blue diamonds indicate confidence intervals. **b–e** Kaplan-Meier survival analysis for SCLC patients with high or low expression levels of PPAT (**b**), PRPS2 (**c**), UMPS (**d**), or GLS1 (**e**) genes in their tumors (GSE60052). Patients were divided according to the median gene expression level (Two-sided, $P \leq 0.05$). **f–k** Immunoblot analysis of PPAT in MS-1-L (**f**), STC-1 (**h**), and SBC-3 (**j**) SCLC cell lines stably infected with retroviruses encoding luciferase control (Luci-KD) or two independent PPAT (PPAT-KD#2 or -#3) shRNAs, was conducted with single time. The intensity of PPAT bands (indicated by the arrowheads) was measured by densitometry ($n = 1$). In **j**, PPAT-depleted SBC-3 cells were also infected with a retrovirus for wild-type (WT) human PPAT. The proliferation rate in 3-D culture for the infected MS-1-L, STC-1, and SBC-3 cell lines was also determined ($n = 5$ [**g** and **i**] or 12 [**k**]). **l** Immunoblot analysis of GLS1 in SBC-3 cells stably overexpressing (OE) wild-type human GLS1 or infected with the corresponding empty retrovirus, was conducted with single time. **m** Proliferation rate in 3-D culture ($n = 6$) for cells as in **l**. In **g**, **i**, **m**, colonies were counted in two randomly selected fields in each of three dishes ($n = 6$). In **k**, colonies were counted in four randomly selected fields in each of three dishes ($n = 12$). Quantitative data in **g**, **i**, **k**, and **m** are means + s.d. *$P < 0.05$, **$P < 0.01$, ***$P < 0.001$, ****$P < 0.0001$ (paired two-tailed Student's $t$-test [**a**, **g**, **i**, **k**, **m**] or log-rank test [**b–e**]). Source data are provided as a Source Data file.

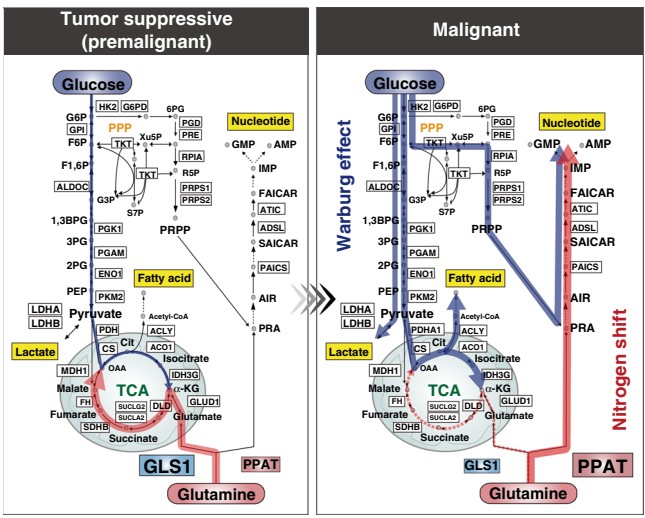

**Fig. 8 The fate of nitrogen in glutamine is shifted in a model of malignant progression.** Proteomic and metabolomic analyses unveil the entire landscape of metabolic changes in a model of malignant cancer. In addition to the carbon shift of the Warburg effect, a nitrogen shift plays a key role in malignant progression in the model. PPAT and GLS1 are decisive factors controlling the nitrogen shift from glutamine.

5′-AACGTTTCAGTCTGAAAGAGAA-3′ and 5′-CACGATCTTGTTTCTCTG TGTA-3′ for GLS1 shRNAs #2 and #3, respectively; and 5′-GCCTGAAG TCTCTGATT-3′ for luciferase shRNA. When the specific shRNA used is not indicated, the experiment was performed with PPAT or GLS1 shRNA #3.

**Protein overexpression.** Complementary DNAs encoding wild-type human PPAT or wild-type or S286A mutant forms of human GLS1 were subcloned into pCX4.

**Malignant progression model.** A single-cell suspension of TSM cells ($4 \times 10^4$) in 3 ml of DMEM supplemented with 10% FBS and 2.5% methylcellulose (Wako) was layered on top of the same medium containing 0.5% agar in a 60-mm culture dish. After culture for 10 days, the formed colonies were picked up and expanded by monolayer culture in DMEM supplemented with 10% FBS and antibiotics. This procedure was repeated a total of three times to yield AIG-1, AIG-2, and AIG-3 cells, consecutively.

**Colony formation assay.** A single-cell suspension of TIG-3–derived cell lines ($1 \times 10^5$ cells) in 3 ml of DMEM supplemented with 10% FBS and 0.33% agar was layered on top of the same medium containing 0.5% agar in a 60-mm culture dish. Colonies formed after culture for 16 days were stained with 2-(4-iodophenyl)-3-(4-nitrophenyl)-5-phenyl-2$H$-tetrazolium chloride (INT, Dojindo) and photographed. A single-cell suspension of A549, HeLa, or other human cancer cell lines ($2 \times 10^3$ to $1 \times 10^4$ cells) in 2 ml of DMEM supplemented with 10% FBS and 0.33% agar was layered on top of the same medium containing 0.5% agar in each well of a six-well plate. Colonies formed after culture for 20 days were stained with INT. Colonies were counted in two randomly selected fields in each of three dishes or wells.

**Immunoblot analysis.** Cell lysis and immunoblot analysis were performed[60]. Immunoblot signals were quantified with the use of an ImageQuant LAS-4000 instrument and ImageQuant TL software (GE Healthcare Life Sciences).

**Antibodies.** Antibodies to c-Myc (1:1000, ab32072) and to GLS1 (1:1000, ab60709; or ab93434 in Supplementary Figs. 7e and 10a) were obtained from Abcam, those to PPAT (1: 2000, 15401-1AP) and PDHA1 (1:1000, 18068-1-AP) were from Proteintech, those to Hsp90 (1:5000, 610419) were from BD Transduction Laboratories, and those to GAPDH (1:1000, ENZ ADI-CS) were from Enzo.

**Glutaminase inhibitor.** The GLS1 inhibitor CB-839 was obtained from Cayman Chemical (22038).

**Nucleotides.** AMP (A2252) and GMP (G8377) were obtained from Sigma.

**Mice.** Female athymic nude (Balb/c-nu/nu) mice were obtained from CLEA Japan. All mice were housed with chow and water supplied automatically, under pathogen-free conditions and a 12-h dark/light cycle. All mouse experiments were approved by the Animal Ethics Committee of Kyushu University.

**Tumorigenicity assay.** TIG-3(TSM) or AIG-3 cells ($2 \times 10^7$), A549 cells ($5 \times 10^6$), or HeLa cells ($2 \times 10^6$) were injected subcutaneously into 6- to 10-week-old female athymic nude mice, after which tumor size was measured once a week. Tumor volume (mm$^3$) was calculated as: (length × width$^2$)/2.

**Sample preparation for iMPAQT.** Cells ($2 \times 10^6$) were washed with ice-cold PBS, exposed to 0.25% trypsin-EDTA (Gibco) for 2 min at 37 °C, and resuspended in DMEM supplemented with 10% FBS. The cells were collected by centrifugation ($800 \times g$ for 5 min), washed twice with PBS, and resuspended in 2–5 ml of PBS, after which the cell number was determined with an automated cell counter (Moxi Z, ORFLO). Portions of the cell suspension were transferred to 1.5-ml tubes and centrifuged ($800 \times g$ for 5 min), and the resulting cell pellets were stored at –80 °C until analysis. The frozen cells ($2 \times 10^6$) were subsequently lysed with 200 µl of a solution containing 2% SDS, 7 M urea, and 100 mM Tris-HCl (pH 8.8); subjected to ultrasonic disruption with a Bioruptor (Diagenode) five times for 30 s, with 30-s intervals between treatments; diluted with an equal volume of water; again subjected to ultrasonic disruption according to the same protocol; and assayed for protein concentration with the bicinchoninic acid (BCA) assay. Portions of each lysate (200 µg of protein) were subjected to methanol-chloroform precipitation to remove detergent and buffer by the sequential addition of 600 µl of ice-cold methanol, 200 µl of chloroform, and 400 µl of water. The samples were mixed for 30 s, allowed to stand for 30 min on ice, and then centrifuged at $21,000 \times g$ for 5 min. The protein pellet was suspended in 1 ml of ice-cold methanol, and the mixture was centrifuged consecutively at $2070 \times g$ for 5 min in a swing-type rotor and at $21,000 \times g$ for 2 min in a fixed-angle rotor (Tomy MX-105). The final pellet was washed twice with ice-cold 80% methanol, dissolved in 28 µl of digestion buffer (0.5 M triethylammonium bicarbonate containing 7 M guanidium hydroxide), incubated at 56 °C for 30 min, and diluted with an equal volume of water. Portions (2 µl) of each sample were then assayed (in triplicate) for protein concentration with the BCA assay. The remaining solution (50 µl) was diluted with 50 µl of water and subjected to digestion with Lys-C (2 µg, Wako) for 3 h at 37 °C. After the addition of 100 µl of water, the samples were further digested with trypsin (2 µg) for 14 h at 37 °C. Cysteine and cystine residues were blocked by treatment of the digest with 5 mM tris(2-carboxyethyl)phosphine for 30 min at 37 °C followed by alkylation with 12.5 mM iodoacetamide for 30 min at room temperature and quenching with 5 mM N-acetylcysteine for 30 min at room temperature. The resulting cell digest was freeze-dried and labeled with the mTRAQΔ0 reagent (1 U, SCIEX) for 2 h at room temperature. Copy number of each enzyme per cell was calculated taking into account the protein amount per cell estimated from the results of the BCA assay.

**MRM data analysis.** Peptides for metabolic enzymes were selected from the iMPAQT database and subjected to chemical synthesis (Supplementary Data 1). The peptides were labeled with the mTRAQΔ4 reagent and added to the mTRAQΔ0-labled sample digests. The corresponding multiple reaction monitoring (MRM) transitions with the expected retention times were obtained from the iMPAQT database. MRM was performed with a triple-stage quadrupole mass spectrometer (QTRAP6500, SCIEX) coupled to a nanoflow liquid chromatography system (Eksigent nano-LC, SICEX). Endogenous peptide abundance was obtained by multiplication of the ratio of the light and heavy intensities summed for each transition and the known amount of synthetic peptide. Protein abundance was determined as the mean ± s.d. for the different peptides for each protein and replicate cultures ($n = 3$) for each cell line.

**Physiological amino acid medium.** DMEM containing high glucose and sodium pyruvate but lacking amino acids (catalog no. 04833575, Wako) was supplemented with 10% dialyzed FBS and amino acids at the same concentrations as present in human plasma[36].

**Metabolome analysis.** For [$^{13}C_6$]glucose labeling in Fig. 2f–h, cells ($2 \times 10^6$) were plated in a 10-cm dish, washed twice with PBS warmed to 37 °C, and incubated in DMEM containing 17.5 mM [$^{13}C_6$]glucose (Sigma-Aldrich, 389374-5 G)[16]. For [$^{13}C_5/^{15}N_2$]glutamine labeling in Supplementary Figure 3, cells ($2 \times 10^6$) were plated in a 10-cm dish, washed twice with PBS warmed to 37 °C, and incubated in physiological amino acid medium (described above) containing 0.6 mM [$^{13}C_5/^{15}N_2$]glutamine (Sigma-Aldrich, 607983). For [$^{13}C_5/^{15}N_2$]glutamine labeling in Figs. 3, 4, cells ($2 \times 10^6$) were plated in a 10-cm dish, washed twice with PBS warmed to 37 °C, and incubated in DMEM containing 2 mM [$^{13}C_5/^{15}N_2$]glutamine (Sigma-Aldrich, 607983) for 15 min (for analysis of the glutamine anaplerotic pathway into TCA cycle) or 6 h (for analysis of nucleotide biosynthesis or amino acids). After labeling, all cells were lysed by the addition of 1 ml of ice-cold methanol, and the lysate was diluted with 400 µl of chloroform and 100 µl of water and subjected to ultrasonic disruption with a Bioruptor five times for 30 s, with

30-s intervals between treatments. The samples were centrifuged at $16,000 \times g$ for 5 min, the upper phase (700 µl) was collected, and 271 µl of chloroform and 294 µl of water were added before centrifugation again at $16,000 \times g$ for 3 min. Metabolomics analysis was performed either by ion chromatography with a Dionex IonPac AS11-HC-4 µm column (inner diamater, 2 mm; 250 mm; particle size, 4 µm; Thermo Fisher Scientific) coupled to a quadrupole-Orbitrap mass spectrometer (Thermo Fisher Scientific) for anionic metabolites (organic acids and nucleotides) or by liquid chromatography with a Discovery HS F5 column (inner diameter, 2.1 mm; 150 mm; particle size, 3 µm; Merck) coupled to a quadrupole-Orbitrap mass spectrometer (Thermo Fisher Scientific) for cationic metabolites (amino acids) (Supplementary Data 2). Cell labeling, metabolomics analysis, and data processing were performed at the Division of Metabolomics of the Medical Institute of Bioregulation at Kyushu University[61]. "Percent labeled" shown in figures refers to the proportion of atoms in specific metabolites that are derived from stable isotope-labeled glucose or glutamine.

**Meta-analysis**. We searched the public database PROGgene, which compiles cohort studies from public repositories such as GEO, EBI Array Express, and The Cancer Genome Atlas (TCGA). Studies compiled through 10 March 2018 were selected for inclusion in our meta-analysis on the basis of criteria including a study duration of >7 years, with most of the selected studies having a duration of 10–20 years. About 80 cohort studies met the inclusion criteria and were deemed appropriate for entry into the meta-analysis. Twelve of these studies were subsequently excluded because some values were not clear or because of a single study being analyzed multiple times, leaving a final number of ~80 cohort studies entered into the meta-analysis. Microarray or RNA-sequencing data for all metabolic enzymes in various organs are available online at PROGgene (http://watson.compbio.iupui.edu/chirayu/proggene/database/?url = proggene). Data were combined by means of fixed-effects or random-effects models for each organ. For integration of the HR for various organs, data were combined by means of the random-effects model. Meta-analysis was performed with the use of Review Manager 5.3 (The Nordic Cochrane Centre, The Cochrane Collaboration, Copenhagen, Denmark; http://www.cc-ims.net/RevMan).

**Statistical analysis**. Quantitative data are presented as means ± s.d. unless indicated otherwise and were compared between groups with the paired two-tailed Student's t-test. Principal component analysis was performed for the heat map derived from hierarchical clustering of measured metabolic enzymes with the use of JMP version 13. A P-value of <0.05 was considered statistically significant.

**Reporting summary**. Further information on research design is available in the Nature Research Reporting Summary linked to this article.

## Data availability

The mass spectrometry data are available at Jpost under the project: Quantification of metabolic enzymes in TIG-3 TSM and AIG cells, ID: PXD009543 using the Access key: 6165. URL: https://repository.jpostdb.com/preview/130108915ad96146e3c87. Microarray or RNA-sequencing data for all metabolic enzymes in various organs are available online at PROGgene which compiles cohort studies from public repositories such as NCBI GEO datasets for all of the GSE numbers, EBI Array Express, and The Cancer Genome Atlas (TCGA) datasets for all of the TCGA numbers. (http://watson.compbio.iupui.edu/chirayu/proggene/database/?url = proggene). Lung; GSE5843, GSE11969, GSE26939, TCGA-LUAD, GSE11117, GSE14814, TCGA-LUSC, GSE13213, GSE17710, GSE30219, GSE3141, GSE37745, GSE19188, GSE41271, GSE50081, GSE42127. Breast; NKI, TCGA-BRCA, GSE7390, GSE3494_U133A, GSE1456_U133A, GSE37751, GSE42568, GSE10893-GPL887, GSE18229-GPL887, GSE19783-GPL6480, GSE21653, GSE2607-GPL1390, GSE2607-GPL887, GSE3143, GSE48390, GSE6130-GPL1390, GSE6130-GPL887, GSE9893. Brain; GSE7696, GSE13041_U133, GSE13041_U95v2, GSE16011, TGCA-GBM, TGCA-LGG, GSE16581, GSE2817, GSE30074, GSE37418, GSE42669, GSE4271_U133B, GSE4412_U133A. Hematopoietic; GSE12417_U133A, TCGA-AML, GSE16131_U133A, GSE22762_U133A, GSE23501, GSE2658, GSE4475. Neuroendcrine cancer; GSE62564, TCGA-PCPG. Liver; GSE10141, TCGA-LIHC. Pancreas; GSE21501, GSE28735, TCGA-PAAD, GSE50827, GSE57495, GSE71729. Colorectal; GSE28814, GSE17536, GSE17537, TCGA-COAD, GSE16125, GSE24551, GSE28772, GSE41258, GSE29621, GSE38832, GSE39582.

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

## Acknowledgements

We thank T. Akagi for the pCX4 system and vectors for hTERT and the SV40 early region; K. Yanagihara for human cancer cell lines (National Cancer Center, Japan); JCRB Cell Bank for SCLC cell lines; T. Takami and K. Tsunematsu for technical assistance; and R. Wakabayashi for discussion on meta-analysis. This work was supported in part by KAKENHI grants from Japan Society for the Promotion of Science (JSPS) and the Ministry of Education, Culture, Sports, Science, and Technology of Japan to K.I.N. (18H05215 and 25221303) and to M.M. (17K19606, 16H04730, 17H05534, and 17H06011) as well as by the Project for Cancer Research and Therapeutic Evolution (P-CREATE) of the Japan Agency for Medical Research and Development (AMED). M.M. is also supported by Core Research for Evolutionary Science and Technology (CREST, JPMJCR15G4) of the Japan Science and Technology Agency (JST).

## Author contributions

M.K., K.O., H.S., S.Y., M.T., Y.I., T.B., C.T., and T.T. designed and performed experiments, analyzed data, and prepared the manuscript. M.M. and K.I.N. contributed to supervision of the study and writing the manuscript.

## Competing interests

The authors declare no competing interests.
