## [Peer Review File · Nature Communications]

Reviewers' comments:

Reviewer #1 (Remarks to the Author):

The authors performed a revision of the manuscript with additional experiments. This is highly appreciated. However, two major points remain that are essential to be addressed before publication.

1) The definition of glutaminolysis given by the authors is still wrong. Glutaminolysis requires that glutamine carbons to go to AcCoA. What the authors monitor and describe is glutamine anaplerosis. This needs to be corrected!

2) The authors' argument that nucleotides inhibit PPAT does not limit an experiment where nucleotides are given to PPAT KD. Thus, this experiment (nucleotide or nucleoside supplementation to PPAT KS) needs to be done.

Reviewer #2 (Remarks to the Author):

The authors have comprehensively addressed my major critiques. As such, I am generally satisfied with the manuscript and think it would be appropriate for publication. I am particularly intrigued by the data indicating that CB839 can improve the growth of some cells.

One issue that I think is important to raise is that the authors describe pyruvate dehydrogenase activity as being anaplerotic (lines 139-140). Whereas pyruvate carboxylase activity is anaplerotic, pyruvate dehydrogenase activity is explicitly not anaplerotic. Anaplerosis is generally defined as replenishing 4 carbon backbones into the TCA cycle, so adding acetyl-coa to oxaloacetate does not fulfill this function. This mistake has been made before (<https://www.nature.com/articles/nature14624>) and it is important to not further contribute to this confusion.

Reviewer #3 (Remarks to the Author):

Kodoma et al. have addressed many of the concerns that arose in the initial draft of the manuscript. In general, the revised version is improved in both scientific rigor and interpretation of results. The authors may consider acknowledging subcellular distribution of Gln-derived Glu (cytosolic vs. mitochondrial) as an important open question.

Response to Reviewer #1

We thank the reviewer for the careful evaluation of our manuscript and for the statement that “The authors performed a revision of the manuscript with additional experiments. This is highly appreciated.” We also thank the reviewer for several suggestions that we feel have helped us to improve our manuscript. Our specific responses to the points raised are as follows:

1. *The definition of glutaminolysis given by the authors is still wrong. Glutaminolysis requires that glutamine carbons go to AcCoA. What the authors monitor and describe is glutamine anaplerosis. This needs to be corrected!*

[Response] We apologize for the inadequate description of glutaminolysis. We have now adequately rephrased the word throughout the revised manuscript (page 2, lines 26–27; page 3, line 62; page 4, lines 87–88; page 5, lines 131–132, 146–147; page 6, line 162; page 7, lines 201, 205, 208; page 9, line 285; page 11, line 351; page 12, line 373; page 14, lines 446, 450, page 15, line 472).

2. *The authors argument that nucleotides inhibit PPAT does not limit an experiment where nucleotides are given to PPAT KD. Thus, this experiment (nucleotide or nucleoside supplementation to PPAT KS) needs to be done.*

[Response] As suggested by the reviewer, we performed nucleotide supplementation to AIG-3 PPAT-KD cells and examined the effect on growth capacity of the cell. We found that the supplementation of AMP or GMP to PPAT-depleted AIG-3 cells did not improve their capacity for growth (**new Supplementary Fig. 6a**). Given that many nucleotides including IMP, AMP, and GMP were shown to exert feedback inhibition on PPAT and other enzymes in the *de novo* nucleotide biosynthesis pathway (Yamaoka *et al.*, *J. Biol. Chem.* 276, 21285–21291, 2001; Hendstrom, *Chem. Rev.* 109, 2903–2928, 2009), we speculate that the simple supplementation of the nucleotide cannot rescue the growth inhibition of AIG-3 PPAT-KD cells. In addition, exposure of cancer cells to excess AMP inhibits glucose metabolic flux (Mazurek *et al.*, *J. Biol. Chem.* 272, 4941–4952, 1997), making it difficult to interpret these results obtained in the experiments. We have now addressed these points in the revised manuscript (page 8, line 255–page 9, line 259).

Kodama *et al.* Supplementary Figure 6

Response to Reviewer #2

We thank the reviewer for the careful evaluation of our manuscript and for the statements that “I am generally satisfied with the manuscript and think it would be appropriate for publication.” We also thank the reviewer for a suggestion that we feel have helped us to improve our manuscript. Our specific response to the point raised are as follows:

1. One issue that I think is important to raise is that the authors describe pyruvate dehydrogenase activity as be anaplerotic (lines 139-140). Whereas pyruvate carboxylase activity is anaplerotic, pyruvate dehydrogenase activity is explicitly not anaplerotic. Anaplerosis is generally defined as replenishing 4 carbon backbones into the TCA cycle, so adding acetyl-coa to oxaloacetate does not fulfill this function. This mistake has been made before (<https://www.nature.com/articles/nature14624>) and it is important to not further contribute to this confusion.

[Response] We thank the reviewer for the careful evaluation of our manuscript and apologize for the inadequate description of pyruvate dehydrogenase activity. We have now corrected the point in the revised manuscript (page 5, line 141).

REVIEWERS' COMMENTS:

Reviewer #1 (Remarks to the Author):

The authors addressed all my questions.